# SCALING OPTIMAL LR ACROSS TOKEN HORIZONS

**Johan Bjorck**[2,†,*]   **Alon Benhaim**[1,*]   **Vishrav Chaudhary**[3,†]   **Furu Wei**[1]   **Xia Song** [1]
[1]Microsoft    [2] Nvidia    [3] Meta

## ABSTRACT

State-of-the-art LLMs are powered by scaling – scaling model size, training to-kens, and cluster size. It is economically infeasible to extensively tune hyper-parameters for the largest runs. Instead, approximately optimal hyperparame-ters must be inferred or *transferred* from smaller experiments. Hyperparameter transfer across model sizes has been studied in Yang et al. (2022). However, hy-perparameter transfer across training tokens – or token horizon – has not been studied yet. To remedy this we conduct a large-scale empirical study on how op-timal learning rate (LR) depends on the token horizon in LLM training. We first demonstrate that the optimal LR changes significantly with token horizon – longer training necessitates smaller LR. Secondly, we demonstrate that the optimal LR follows a scaling law and that the optimal LR for longer horizons can be accurately estimated from shorter horizons via such scaling laws. We also provide a rule-of-thumb for transferring LR across token horizons with zero overhead over current practices. Lastly, we provide evidence that LLama-1 used too high LR, and thus argue that hyperparameter transfer across data size is an overlooked component of LLM training.

## 1 INTRODUCTION

State-of-the-art LLMs are scaled in multiple dimensions. The models are becoming increasingly large, e.g. Grok-1.5 has 314 billion (B) parameters (xAI, 2024). The clusters used to train them are growing in size, e.g. the recently operational Memphis super-cluster contains over 100,000 H100 GPUs (Alcorn, 2024). Lastly, the training datasets are growing, e.g. LLama-3 was trained on 15 Trillion (T) tokens (Dubey et al., 2024). At these scales, it is infeasible to extensively tune hyperparameters. Practitioners must instead resort to *hyperparameter transfer*, a process where approximately optimal hyperparameters for large-scale experiments are inferred from experiments at a smaller scale. Perhaps the most famous work on hyperparameter transfer is muP (Yang et al., 2022) – a methodology for transferring optimal hyperparameter from a small model to a large model.

---

[*] Equal contribution. [†]Work done while at Microsoft.

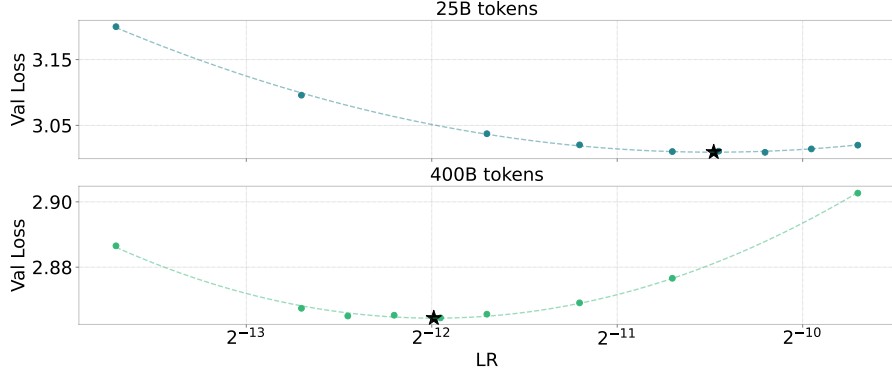

Figure 1: Final validation loss of a 350 million parameter LLM for different learning rates (LR) and token horizons. The dashed lines indicate our fitted curve and the stars indicate the estimated optimal LR. The optimal LR decreases as the token horizon increases.

While hyperparameter transfer across model size is a well-studied problem, transfer across token horizon remains understudied. This paper aims to remedy this shortcoming in the literature.

In this paper, we present a large-scale study on hyperparameter transfer across token horizons, we define the latter as the total number of tokens seen during training. We specifically focus on learning rate (LR), an important hyperparameter that influences training stability, generalization, and convergence speed. Our study is essentially a large ablation experiment where we vary LR and token horizon for a few different LLM models. We consider >250 training runs in total. To keep the scope and computational requirements manageable we focus on how LR depends on the token horizon, and only present some preliminary results on its interaction on model size. Our experiments are based on standard public resources – the Megatron codebase (Shoeybi et al., 2019) and the RefinedWeb dataset (Penedo et al., 2023). Our three main contributions are:

- We provide a large-scale study demonstrating that the optimal LR depends strongly on the token horizon, with longer horizons necessitating smaller LRs. This fact holds even when muP parametrization is used.

- We demonstrate that 1) the optimal LR for any given architecture follows a scaling law and 2) this allows *hyperparameter transfer* where the optimal LR for a long horizon is inferred from a shorter horizon. Furthermore, we provide a rule-of-thumb for transferring LR across token horizons with zero overhead over current practices.

- We provide a case study on the optimal LR for the LLama architecture. We provide evidence that LLama-1 used an LR that is too large, highlighting hyperparameter transfer across horizons as an overlooked component of LLM training.

## 2 BACKGROUND

**LLM Scaling Laws.** LLMs are typically decoder-only transformers (Vaswani, 2017) trained via next token prediction on web-scale text (Radford et al., 2019). It has been empirically observed that the performance of LLMs scale well with model size – with the largest model showing emergent capabilities (Wei et al., 2022). Kaplan et al. (2020) shows that LLM performance roughly scales as a **power-law** in the model size $N$. The performance here is measured by validation loss $L$, which is well known to correlate strongly with downstream metrics. Specifically they propose the following law (using constants $N_c, \alpha_N$) for models trained on sufficiently large datasets: $L(N) = (N_c/N)^{\alpha_N}$. Hoffmann et al. (2022) also showed that the performance scales well with *token horizon* and that the optimal performance for a given FLOPs budget is obtained by scaling the model and token horizon jointly. The current paradigm thus scales token count in addition to model size – e.g. LLama-3 was trained on >10x as many tokens as LLama-1 (Dubey et al., 2024). In this paper we use notation from Kaplan et al. (2020), denoting the token horizon (i.e. number of tokens seen during training, some of which might be duplicate) by $D$ and the model size (i.e. parameter count) by $N$. As is common in the literature, we will fit scaling laws to empirical observations. Following Kaplan et al. (2020), we will use power-laws of the form $F(X) = AX^\alpha$, where $\alpha$ and $A$ are fitted and $X$ is the quantity of interest. To measure goodness-of-fit we will use the $R^2$ measure – its value ranges from 0 to 1, where 1 indicates a perfect fit and 0 indicates no fit.

**Hyperparameter transfer.** Hyperparameters can strongly influence the performance of LLMs, and LR is a critical hyperparameter. The muP paper (Yang et al., 2022) first popularized *hyperparameter transfer* – the process of finding optimal hyperparameters from a small proxy experiment. Core to muP is the muP-parametrization – a way of parametrizing an LLM such that the optimal LR for a small model is also optimal for a larger model. The muP parametrization introduces some changes to the network, e.g. the attention scaling factor is changed from $\frac{1}{\sqrt{d}}$ to $\frac{1}{d}$. The original muP paper shows that LR and many other hyperparameters transfer from small to large models when muP parametrization is used, which is further corroborated in Lingle (2024). While Yang et al. (2022) focuses on transfer between model sizes, we focus on transfer between token horizons.

## 3 EXPERIMENTS

**Experimental Setup.** Our experimental setup closely follows the setup of the GPT-3 paper (Brown, 2020). We consider model sizes of 50 million (m), 125m, 350m, 760m, 1.3 billion (B), and 2.7B

parameters using the architectures of Table 2.1 in the GPT-3 paper. The table is replicated as Table 4 in Appendix A and also lists the LRs used in the GPT-3 paper. We use hyperparameters following GPT-3 – weight decay of 0.1, gradient clipping of 1.0, and cosine learning decay schedule. The full list of hyperparameters can be viewed in Table 3 in Appendix A. The LR will vary with the training step since we use an LR schedule, and we will use the term LR to refer to the largest LR of a run. We make three adjustments compared to the GPT-3, mainly to prevent divergence of the model for high LRs. 1) For warmup we use the maxima of 1% of the training steps following Muennighoff et al. (2024) and 1000. Having too short a warmup stage is known to cause divergence (Wortsman et al., 2023) which we want to avoid for short training runs. 2) We use qk-norm following Dehghani et al. (2023), this is known to prevent divergence without hurting validation loss (Wortsman et al., 2023). 3) The original GPT-3 paper uses different batch sizes for different model sizes. To avoid confounders we simply use the same batch size of 0.5m tokens all model. We use the RefinedWeb dataset (Penedo et al., 2023), a common-crawl derived dataset of roughly 600B tokens which is known to be of high quality (Penedo et al., 2024). Experiments are run on the Megatron codebase (Shoeybi et al., 2019). Unless mentioned we will use the same seed for all runs. For curve fitting we use least-square fitting with either a first or second-degree polynomial using Numpy and Scipy (Harris et al., 2020), and we always have more data points than free parameters in the fits.

## 3.1 ABLATIONS

In our first experiment, we consider the 350m LLM model from Table 4 in Appendix A. We perform an ablation study where we vary the LR and token horizon and measure the final validation loss. We consider $\{25, 50, 100, 200, 400\}$ billion tokens. We will start with the LR of $3 \times 10^{-4}$ from Table 4 which is what the GPT-3 paper used (Brown, 2020). We then multiply this base LR by factors $\{0.25, 0.5, 1, 2, 4\}$. The LRs we consider are thus $\{7.5 \times 10^{-5}, 1.5 \times 10^{-4}, 3 \times 10^{-4}, 6 \times 10^{-4}, 1.2 \times 10^{-3}\}$. For each combination of LR and token horizon, we train a model and record its final validation loss. To make sure we have the best possible resolution around the minima we find the optimal learning rate $LR^*$, and then further train with the LRs halfway between $LR^*$ and its two nearest neighbors – a procedure we repeat twice. From these losses, we fit a curve that estimates how the final validation loss depends on the LR. For each token horizon we fit a second-degree polynomial in $\log(LR)$, using a quadratic polynomial as it provides an excellent fit and is the simplest polynomial with a well-defined minimum. The $R^2$ of the fit are 0.995 or better, see Table 6

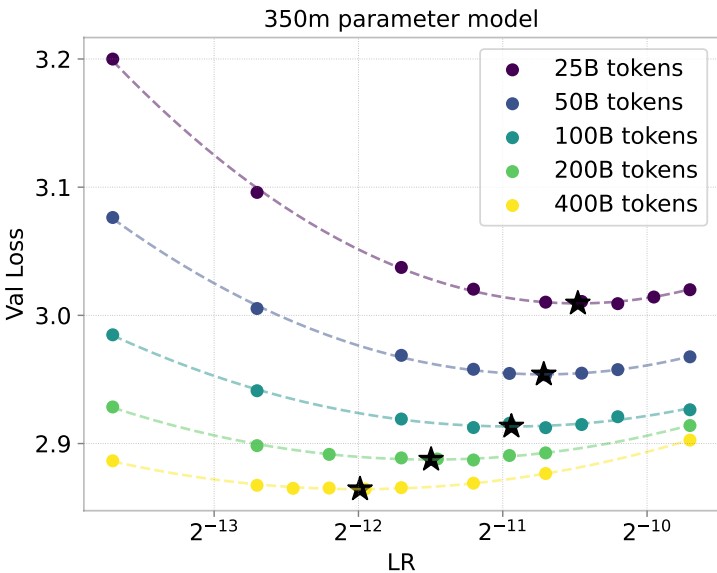

Figure 2: Final validation loss as a function of learning rate (LR) and token horizon. The dashed lines indicate our fitted curve and the stars indicate optimal LR. The optimal LR decreases monotonically with longer horizons.

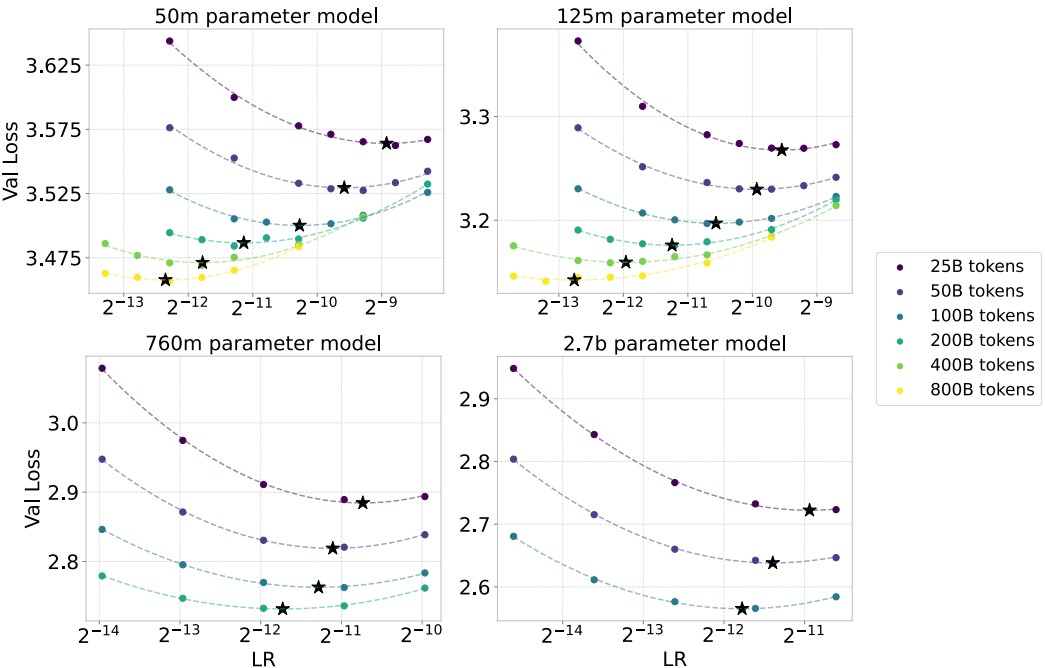

Figure 3: Final validation loss as a function of max learning rate (LR) and token horizon for four models. The dashed lines indicate our fitted curve. The optimal LR, denoted by a black star, decreases monotonically with longer horizons for all models.

in the Appendix for exact values. The validation losses and fitted curve are shown in Figure 2. We also estimate the optimal LR by taking the minimizer of the fitted curve. From Figure 2 we make two observations: 1) **the optimal LR decreases with longer token horizons**, and 2) the quadratic fit provides excellent agreement with experimental data.

We now repeat these experiments for more model sizes – specifically the 50m, 125m, 760m, 1.3B, and 2.7B parameter models from Table 4. For computational reasons, we only consider shorter horizons for the larger models. The 50m and 125m models go up to 800B tokens, the 760m and 1.3B models go up to 200B tokens while the 2.7B model only goes up to 100B tokens. For the larger models, we don't increase the sampling rate around the minimizer. For each model, we consider the base LR as the one used in the GPT-3 paper (also listed in Table 4) and then multiply it by e.g. $\{0.25, 0.5, 1, 2, 4\}$. The results of these experiments are shown in Figure 3. Two runs (model sizes 50m and 125m, 800B tokens, and highest LR) diverge, and we remove these. The results look similar to those of Figure 2 and we observe that **the optimal LR decreases with longer token horizons across model sizes**. Figure 10 in the Appendix show similar results for the 1.3B model. We thus conclude that this is a robust phenomenon.

## 3.2 SCALING LAWS

We have seen that longer token horizons require smaller LR. We now investigate if this insight allows us to derive scaling laws and do *hyperaprameter-transfer* – i.e. finding the optimal LR to a long token horizon from experiments on a shorter horizon. To do this we will fit scaling laws to our empirical results. Given some fixed model architecture and training recipe, let $LR^*(D)$ denote the optimal LR for some token horizon $D$. We will use the following functional form:

$$LR^*(D) = BD^{-\beta} \tag{1}$$

Here $B$ and $\beta$ are two constants independent of $D$ that might e.g. depend on the model architecture. Taking the logarithm of both sides of Equation (1) we get

$$\log\left[LR^*(D)\right] = \log B - \beta \log D \tag{2}$$

We thus have a linear equation in the unknowns $\log B, \beta$, and can fit these to our experimental results with least squares. Specifically, we use the minimizer of the quadratic fits of Figure 3 as the data

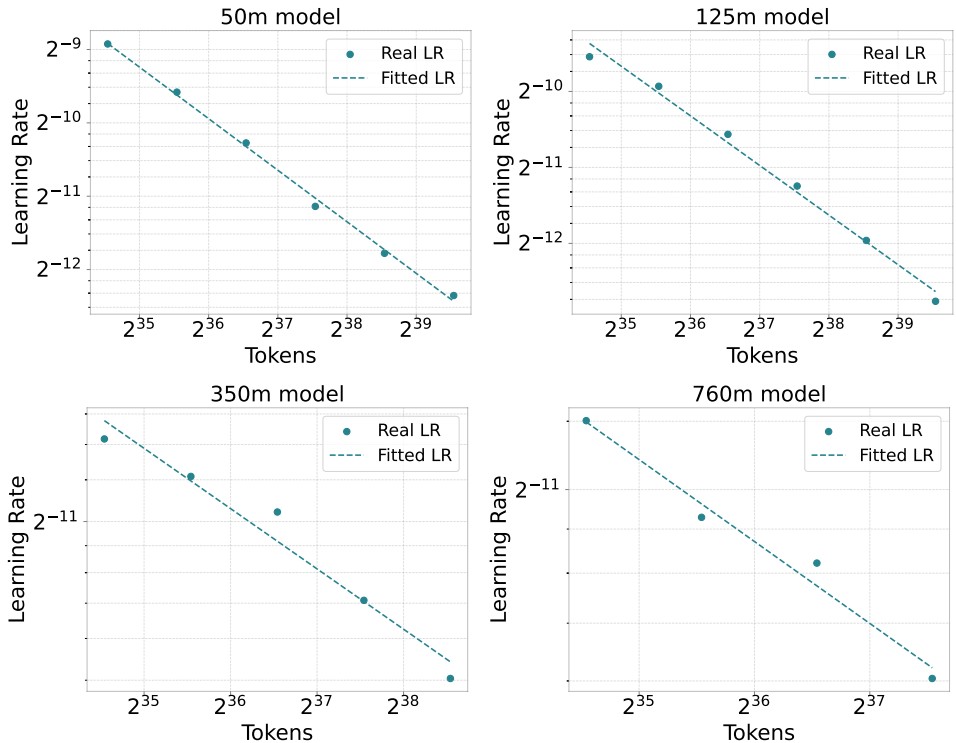

Figure 4: Scaling laws for optimal LR versus token Horizon. We compare the empirically best LR (dots) to the smooth scaling law of Equation (1) with fitted constants. The $R^2$ of these fits are in the range 0.99 - 0.96. Across all model sizes, we see that the scaling law provides a good fit to the empirical data.

points $LR^*$. Fitting Equation (2) gives good fits – the curves for four models are shown in Figure 4. The $R^2$ of these fits are in the range 0.99-0.96 (see Table 5 in Appendix A for exact values), a rather high number. In Figure 13 in the Appendix we also show the fits for the 1.3B and 2.7B models, which also show a good fit. In Figure 11 in the Appendix we repeat these experiments on a small scale by using the Llama architecture on the smallest 50m model, and find that the $\beta$ transfer well across architectures when using the same model size. In Figure 12 we repeat these experiments by doing multiple epochs over a fixed dataset of 25B tokens, and observe similar scaling behavior.

To *evaluate* the predictive power of the scaling laws we cannot solely rely on $R^2$ as that is essentially a measure of training loss. We instead need to evaluate the fit on some held-out data. To simulate hyperparameter transfer we thus consider fitting the constants $\log B, \beta$ on token horizons $25B, 50B$ and $100B$ – and then use these constants to predict the optimal LR at longer horizons. We will use the optimal LR we have empirically found with quadratic fits as the correct LR. The results are illustrated for the 50m model in Table 1. We see a reasonably good fit with a relative error of 10-15% – a clear improvement over not scaling LR at all which has a relative error of $> 250\%$. We thus conclude that **it is possible to perform hyperparameter transfer across token horizons with our scaling laws**. Table 1 is repeated for more models with similar fits in Appendix B.

### 3.3  muP PARAMETRIZATION

It is natural to ask if muP allows hyperparameter transfer across token horizons. To investigate this we consider the 50m model from Table 4, and use muP parameterization. Note, we will not perform hyperparameter transfer across model sizes, we just use the muP parameterization which slightly differs from the standard parameterization of transformers. We then run ablation experiments using token horizons $\{25, 50, 100\}$ billion tokens and LRs of $\{0.25, 0.5, 1, 2, 4\}$ times the base LR in Table 4. The results of this experiment are shown in Figure 5, and we can indeed see that the optimal LR decreases with a longer token horizon. We thus conclude that **the optimal LR does not transfer across token horizons with muP**.

|  | 25B | 50B | 100B | 200B | 400B | 800B |
|---|---|---|---|---|---|---|
| Optimal LR | $1.54\times10^{-3}$ | $9.79\times10^{-4}$ | $6.06\times10^{-4}$ | $3.33\times10^{-4}$ | $2.14\times10^{-4}$ | $1.71\times10^{-4}$ |
| Predicted LR |  |  |  | $3.81\times10^{-4}$ | $2.39\times10^{-4}$ | $1.50\times10^{-4}$ |
| Ratio |  |  |  | 0.873 | 0.894 | 1.14 |

Table 1: Simulated hyperparameter transfer across token horizons for a 50m model. We use the empirically measured optimal LR at 25,50 and 100B tokens to estimate the optimal LR at longer horizons by fitting the constants of Equation (2). We find a reasonable fit when scaling up the horizon to 800B tokens, with a relative error of 10-15 %. The relative error of using the best LR at the 100B horizon for an 800B horizon is $> 250\%$. The scaling laws thus have predictive power.

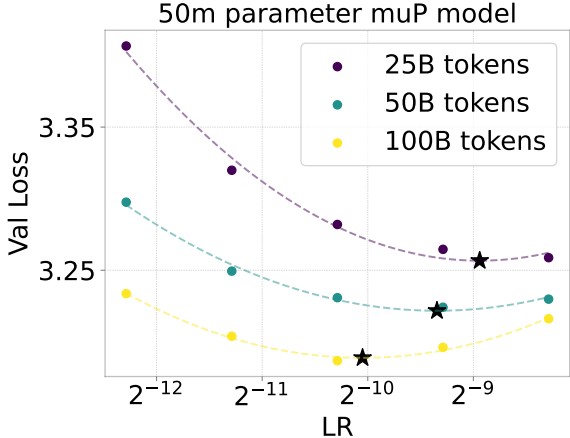

Figure 5: Optimal LR vs token horizon for a 50m model using muP parameterization (Yang et al., 2022). We see that the optimal LR decreases with longer token horizons, demonstrating that LR does not transfer across horizons even with muP.

### 3.4 QUANTIFYING VARIANCE

To ensure that our results are reliable we here consider quantifying the variance of our experiments. We will use two different methodologies. Firstly we use bootstrapping, following Hoffmann et al. (2022). We consider the 350m model of Table 4, randomly remove 20% of the data points, and then fit the optimal LR at each token horizon and the scaling law of Equation (1). We repeat this procedure 1000 times and then measure the mean and std of the optimal LR and the constants of Equation (1). The results are given in Table 2. We see that the standard deviations are relatively small, which again suggests that the variance in our experimental results is low.

The second methodology we use is simply rerunning a small-scale experiment with multiple seeds. We use the 350m model from Table 4, a 100B token horizon, three learning rates, and two additional seeds. We then estimate the optimal LR via a quadratic fit as in Section 3.1. The results of this experiment are shown in Table 7 in Appendix B. We see that there are small differences in the losses, and hence small differences in the estimated optimal LR. The relative std $\sigma/\mu$ is $2.63\times10^{-2}$, so we can expect the relative error to be a few percent. When using more data points the variance could further decrease due to concentration of measure.

### 3.5 EFFECT OF BATCH SIZE

It is well known that batch size $BS$ affects the optimal LR (Goyal, 2017). While we primarily focus on the setting of a fixed batch size, we here consider modifying the batch size for the 1.3B model of Table 4. Specifically, we double the batch size to 1m tokens and train for 25B and 50B tokens in total with different LRs. In Figure 9 in the Appendix we show the final validation loss as a function of the LR and estimate the optimal LR. In Figure 15 we show the optimal LR as a function of token horizon for the 1.3B model using a batch size of 0.5 or 1m tokens. We see that the optimal LR is

|  | 25B | 50B | 100B | 200B | 400.0 |
|---|---|---|---|---|---|
| $\mu(LR^*)$ | $7.04{\times}10^{-4}$ | $5.96{\times}10^{-4}$ | $5.07{\times}10^{-4}$ | $3.45{\times}10^{-4}$ | $2.44{\times}10^{-4}$ |
| $\sigma(LR^*)$ | $1.28{\times}10^{-5}$ | $9.12{\times}10^{-6}$ | $1.96{\times}10^{-5}$ | $4.07{\times}10^{-6}$ | $4.31{\times}10^{-6}$ |
| $\sigma/\mu$ | $1.82{\times}10^{-2}$ | $1.53{\times}10^{-2}$ | $3.88{\times}10^{-2}$ | $1.18{\times}10^{-2}$ | $1.76{\times}10^{-2}$ |

Table 2: We estimate the mean $\mu$ and standard deviation $\sigma$ of the optimal learning rate $LR^*$ via bootstrapping. We sample 80% of the data from Figure 2 and then estimate the optimal LR with the procedure from Section 3.1. This bootstrapping procedure is repeated 1000 times. We see that the variance is small compared to the mean, and the relative error $\sigma/\mu$ is on the order of a few percent. This implies that the uncertainty in our estimates is small.

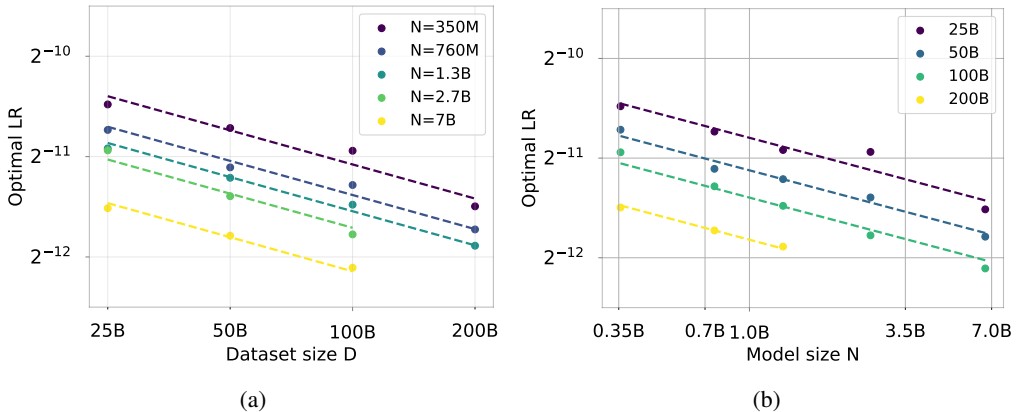

(a)               (b)

Figure 6: **(a)** There is a linear relationship between $\log D$ and $\log LR^*$ for different model sizes $N$. **(b)** There is a linear relationship between $\log N$ and $\log LR^*$ for different values of dataset size $D$.

higher with a larger batch size as expected. More importantly, we note that the linear fits are roughly parallel. This suggests that the optimal LR depends on the token horizon the same way irrespective of the batch size, i.e. that we can factorize Equation (1) as $LR(BS, D) = f(BS)D^{-\beta}$.

## 4 SCALING LAW WITH RESPECT TO MODEL SIZE

So far we have considered Equation (1) with constants fitted independently for each model size. Fully determining the joint scaling properties of the model architecture and token horizon is outside of the scope of this paper – doing so would be computationally infeasible. Nonetheless, we will here provide some preliminary results and discussions regarding a scaling law for both model size $N$ and token horizon $D$. In Figure 6b we plot the optimal LR as a function of $N$, and in Figure 6a we plot it as a function of $D$. Both curves are straight lines when the axes are logarithmic. This observation motivates the following functional form, which is mathematically derived in Appendix C.

$$LR^*(N, D) = CN^{-\alpha}D^{-\beta} \tag{3}$$

Here, the constant $C$ is the ideal learning rate for a model of size $1B$ and token horizon $1B$, where the precise numerical value of $C$ may depend on the batch size (see section 3.5), vocabulary size and tokenization. The factor $N^{-\alpha}$ captures the fact that the optimal LR decreases with model size $N$, while the factor $D^{-\beta}$ captures that optimal LR also decreases with token horizon $D$. We now fit Equation (3) to our data for the models of size 760m, 1.3B, and 2.7B, using the 7B model of Section 4.1 as a held-out validation set. We use the huber loss with $\delta = 1 \times 10^{-3}$ and the BFGS algorithm, similar to Hoffmann et al. (2022). We observe a good fit with RMSE$= 2.2 \times 10^{-5}$ and a validation $R^2$ of 0.978. The numerical values we find are:

$$C \sim 1.55 \times 10^{-3}, \quad \alpha \sim 0.23, \quad \beta \sim 0.32 \tag{4}$$

We depict the results in Figure 7. For smaller models (mainly 50m and 125m) a larger $\beta$ is observed from the experiments. We thus consider Equation (3) to be valid in the regime of large models ($\geq$ 760M parameters).

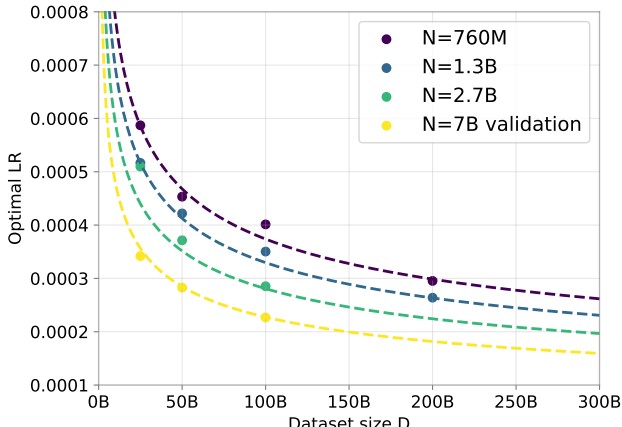

Figure 7: Fit of Equation (3) compared to the experimental results. The data points for the 7B model of Section 4.1 are excluded at the time of fitting and used as validation data. We have an $R^2$ of $0.978$ on this validation data. Note, the 7B model uses the Llama architecture while the other data points use the GPT-3 architecture. This experiment thus demonstrates that the scaling laws have predictive power across model architectures.

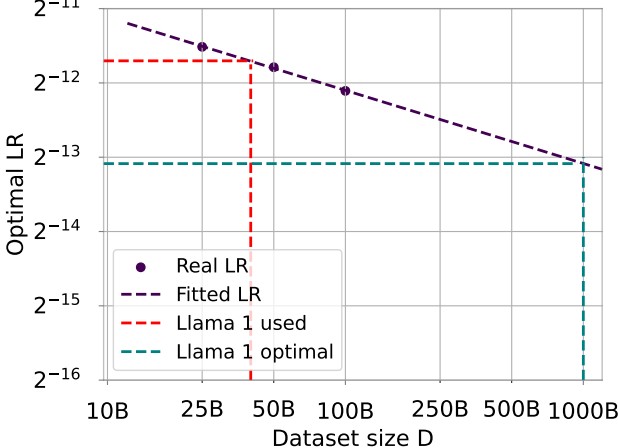

Figure 8: The optimal LR of the LLama-1 7B model as a function of token horizon $D$. We find that $1.15\times10^{-4}$ is optimal for 1T tokens whereas the Llama-1 paper used $3\times10^{-4}$.

## 4.1 A CASE-STUDY ON LLAMA-1

We now consider evaluating if the LRs used by Llama-1 Touvron et al. (2023) are "correct" according to our scaling laws. To do this we adopt the LLama-1 architecture (RMSnorm, Rope embeddings, and so on) and run small-scale experiments with token horizons 25B, 50B, and 100B and different LRs. Based upon these experiments, we find that values for Equation (1) are $B = 8.29\times10^{-4}$ and $\beta = 0.3$ (see Figure 14 in Appendix B). We then extrapolate to find the optimal LR at 1T tokens, which comes out as $1.15\times10^{-4}$. LLama-1 used $3\times10^{-4}$, an LR which is too large by a factor of $> 2.5$ according to our results. This methodology is visualized in Figure 8. The numerical values used for our predictions are also tabulated in Table 8 in Appendix B, where we also compare to predicting the optimal LR for LLama-1 using our scaling law from Eq. Equation (3). If Llama used the wrong LR, what were the actual effects of this? We provide an estimate on the *upper bound* for the validation loss penalty between the optimal and used learning rate in Llama-1 using the parabola fit at $D = 100B$. $\Delta L = L(D = 100B, LR = LR^*) - L(D = 100B, LR = 2.6LR^*) = 0.027$. Note, this is an upper bound since the parabolas flatten with longer token horizons

## 5 RELATED WORK

The study of scaling laws for LLMs originated in Kaplan et al. (2020) and it is still an active area of research. Scaling laws are researched in the context post-training (Lin et al., 2024; Zhang et al., 2024a; Gao et al., 2023), model architecture (Krajewski et al., 2024; Alabdulmohsin et al., 2024; Frantar et al., 2023; Wang et al., 2023), multi-modality (Aghajanyan et al., 2023; Cherti et al., 2023), inference (Sardana & Frankle, 2023), data (Dohmatob et al., 2024; Fernandes et al., 2023) and other domains (Zhang et al., 2024b; Neumann & Gros, 2022; Kadra et al., 2024). There are also more theoretical studies (Michaud et al., 2024; Caballero et al., 2022).

LR has long been known as an important hyperparameter, and hence there is ample work on how to select it (Goyal, 2017). Larger models are known to require smaller LR, and e.g. Kaplan et al. (2020) suggests the formula $LR(N) \approx 0.003239 - 0.0001395 \log(N)$ for tuning LR as a function of model size. MuP (Yang et al., 2022) is a principled methodology for selecting LR when the model is scaled, and the method is actively researched (Lingle, 2024; Everett et al., 2024; Noci et al., 2024; Blake et al., 2024). Recently Everett et al. (2024) showed that LR transfer across model size is possible both with different parametrizations and optimizers. However, a limitation in both Yang et al. (2022) and Everett et al. (2024) is the fixed training horizon assumption used in the theoretical derivations. Our work directly explores this limitation. One notable conclusion in Everett et al. (2024) is that extrapolating a LR to larger models may significantly overestimate optimal LR in the compute optimal setting. We reach a different conclusion, the exponent for model size is independent of the token horizon and vice versa. While their results might hold true for other optimizers, a closer look at Table 11 in Everett et al. (2024) shows that for the Adam optimizer, the scaling exponents are similar for the two token horizons they considered. Another related paper is Wang & Aitchison (2024), which studies how to set the weight decay of Adam when model and dataset is scaled. At last, we mention that Bi et al. (2024) recently fits the optimal LR as a function of total compute (which will combine batch size, model size, and total training duration) for two small models.

## 6 DISCUSSION

**Limitations.** To limit the scope and computational requirements of our study we have intentionally focused on a narrow area – scaling token horizons and changing LR with otherwise fixed LLM recipes. With this limited scope, there are naturally many limitations to our study. We have only extended the scaling laws to roughly 800B tokens, while many SOTA LLMs are trained significantly longer (Dubey et al., 2024). It is well-known that optimal LR depends on model size, and this study has only scratched the surface of this important topic. Beyond just model size, model architectural modifications like mixture of experts (Shazeer et al., 2017), model width and depth, different attention types (Ainslie et al., 2023), and state-space models (Gu & Dao, 2023) could plausibly interact with both the LR and token horizon. The number of repeated tokens can also play a role, and we only provide small-scale experiments on these theme in Figure 12 in the Appendix. Other dimensions to consider are additional hyperparameters such as weight decay which interacts with LR (Bjorck et al., 2021), LR schedules, and multi-modality (Huang et al., 2023). For computational reasons, we defer these topics to future work.

**Advice for Practioners.** Our experiments show that the optimal LR decreases with the token horizon. This necessitates hyperparameter transfer across token horizons. For practitioners who are working on larger models (say $>= 760m$) we recommend simply using Equation (3) where we have already found $\beta = 0.32$ to generalize across architectures. To find the optimal LR $LR^*(D_1)$ at some long horizon $D_1$ practitioners can just find the optimal LR $LR^*(D_2)$ at a short horizon $D_2$ and then estimate:

$$LR^*(D_2) \approx LR^*(D_1)\left(\frac{D_2}{D_1}\right)^{-0.32} \tag{5}$$

Since practitioners typically find the optimal LR for smaller horizons anyway during hyperparameter search, this methodology has no overhead over current practices. For practitioners with ample compute resources, we recommend finding the best LR at multiple short horizons using the quadratic fitting of Section 3.1. Thereafter the constants in Equation (1) can be found, and the optimal LR at longer horizons can be estimated. For practitioners using methods that provide

zero-shot transfer of optimal LR across width (Yang et al. (2022) and Everett et al. (2024)), we recommend using an adjustment of Equation (3) ($N \propto n_{layer}d_{model}^2$) which yields the formula $LR^*(n_{layer}, D) = Cn_{layer}^{-\alpha}D^{-\beta}$. Then $C$ needs to be found using the usual muP sweep for LR with small width, depth and horizon. This allows hyperparameter transfer of LR across depth.

**Conclusions.** We have investigated how LR and token horizon interact when training LLMs. First, we have shown that the optimal LR decreases as the token horizon gets longer. This finding is robust across model sizes. Secondly, we have shown that the optimal LR follows reliable scaling laws and that fitting these allows for hyperparameter transfer across token horizons. As a case study, we have applied our methods to the training of LLama, and have shown evidence that Llama-1 was trained with a LR which was significantly larger than optimal. We argue thus argue that hyperparameter transfer across token horizons is an understudied aspect of LLM training.

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

# A   HYPERPARAMETERS

| Hyperparameter | Value |
|---|---|
| weight decay | 0.1 |
| grad clip norm | 1.0 |
| LR schedule | cosine |
| Adam $\beta_1$ | 0.9 |
| Adam $\beta_2$ | 0.95 |
| Context length | 2048 |
| Batch size (tokens) | 524288 |
| Warmup Steps | $\max(1000, 0.01 \times \text{train iters})$ |
| Min LR | $0.1 \times$ Max LR |

Table 3: Hyperparameters. These follow Brown (2020). Hyperparameters not listed here follow defaults in the Megatron codebase Shoeybi et al. (2019).

| Model Name | params | layers | $d_{\text{model}}$ | heads | Base LR |
|---|---|---|---|---|---|
| Tiny | 50M | 8 | 512 | 8 | $8 \times 10^{-4}$ |
| Small | 125M | 12 | 768 | 12 | $6 \times 10^{-4}$ |
| Medium | 350M | 24 | 1024 | 16 | $3 \times 10^{-4}$ |
| Large | 760M | 24 | 1536 | 16 | $2.5 \times 10^{-4}$ |
| 1.3B | 1.3B | 24 | 2048 | 16 | $2 \times 10^{-4}$ |
| 2.7B | 2.7B | 32 | 2560 | 32 | $1.6 \times 10^{-4}$ |
| 6.7B | 6.7B | 32 | 4096 | 32 | $1.2 \times 10^{-4}$ |

Table 4: Model architectures and base LRs we consider. These follow GPT-3 Brown (2020).

## B ADDITIONAL EXPERIMENTAL DATA

| Model size | $R^2$ | $\beta$ |
|---|---|---|
| 2.7b | 0.9973 | 0.4184 |
| 1.3b | 0.9898 | 0.3171 |
| 760m | 0.9763 | 0.3155 |
| 350m | 0.9607 | 0.3799 |
| 125m | 0.9895 | 0.6531 |
| 50m | 0.9977 | 0.7029 |

Table 5: The $R^2$ values and found $\beta$s of the fits in Figure 4. The $R^2$s are relatively close to 1, indicating a good fit. The $\beta$s are relatively similar for models $\geq$ 350m, but are larger for the smallest models.

| Token Horizon | $R^2$ |
|---|---|
| 25B | $1- 6.68\times10^{-4}$ |
| 50B | $1 - 7.59\times10^{-4}$ |
| 100B | $1 - 4.68\times10^{-3}$ |
| 200B | $1 - 2.85\times10^{-3}$ |
| 400B | $1 - 1.76\times10^{-3}$ |

Table 6: The $R^2$ values of the fits in Figure 2. They are very close to 1, indicating a great fit.

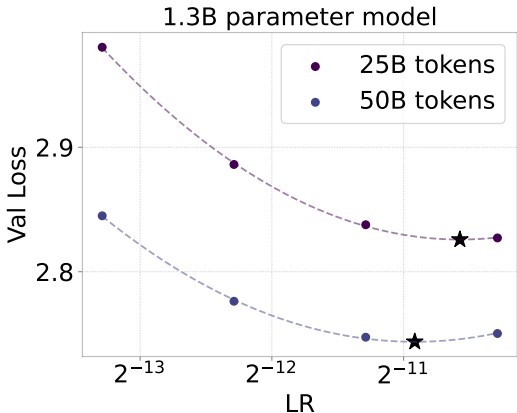

Figure 9: Learning rate and validation loss for different token horizons for the 1.3B model from Table 4 using 1m batch size.

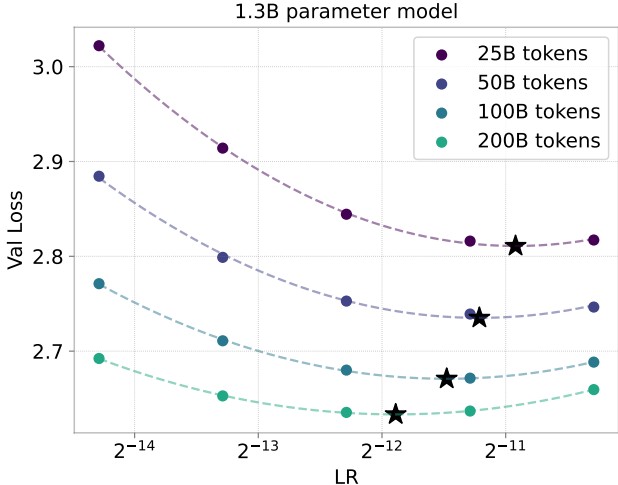

Figure 10: Learning rate and validation loss for different token horizons for the 1.3B model of Table 4. We see that the optimal LR decreases as the token horizon increases.

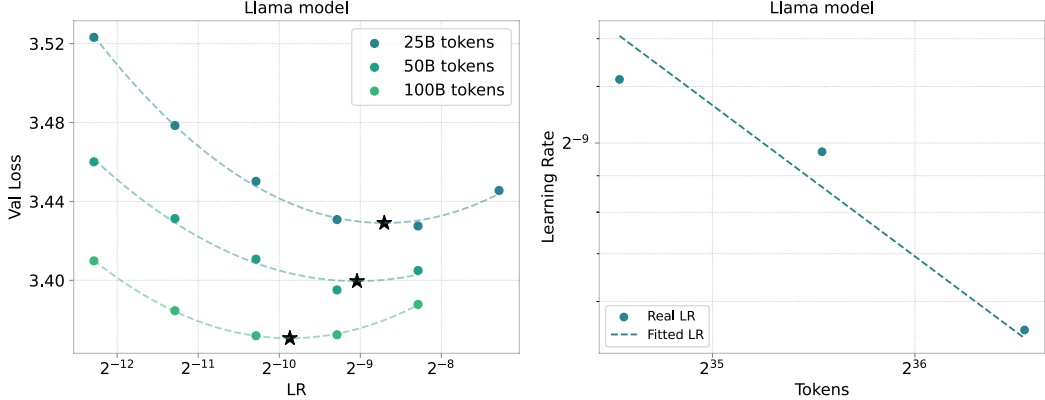

Figure 11: Experiments on the 50m model using the Llama-1 architecture (RMSnorm, Rope embeddings, and so on) of Touvron et al. (2023). Similar to the GPT-3 architecture, we see that optimal LR decreases with the token horizon. On the right we plot the scaling law, using the same $\beta$ as the GPT-3 model. We see an excellent fit, demonstrating that the $\beta$ transfers well across model architectures.

| seed | $L(1.5{\times}10^{-4})$ | $L(3{\times}10^{-4})$ | $L(6{\times}10^{-4})$ | $\arg\min L$ |
|------|------|------|------|------|
| 1 | 2.940372 | 2.919948 | 2.913585 | $5.81{\times}10^{-4}$ |
| 2 | 2.941199 | 2.919131 | 2.912387 | $5.76{\times}10^{-4}$ |
| 3 | 2.941648 | 2.920779 | 2.915190 | $5.47{\times}10^{-4}$ |

Table 7: We consider the loss $L$ as a function of the LR. We repeat the experiments of Section 3.1 with a 100B token horizon, a 350m model, and multiple seeds. With different seeds, we see slightly different final losses, and slightly different estimates of the optimal LR.

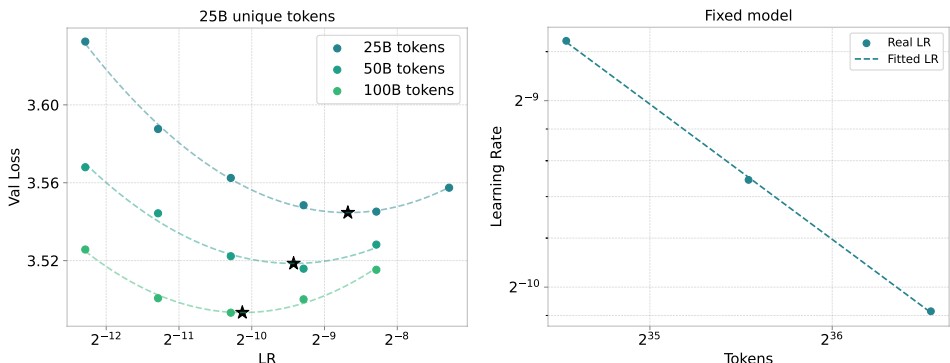

Figure 12: We sample a set of 25B unique tokens, and ablate LR when the training horizon is 25B, 50B and 100B tokens. This corresponds to one, two, or four epochs over the data. We use the 50m model. (Left) The optimal LR decreases when we increase the total token horizon, thus the total number of unique tokens seen does not solely determine the optimal LR. (Right) The slope between the optima in this setting fits the scaling curve from training only only unique tokens. This suggests that it is the token horizon rather than the number of unique tokens which determines the scaling.

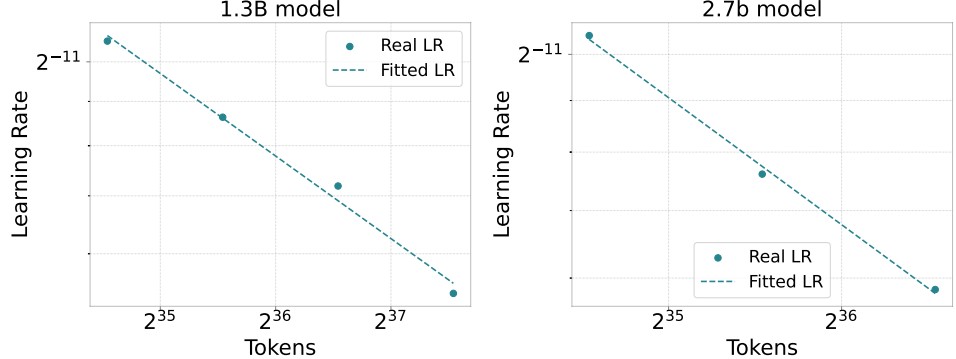

Figure 13: Optimal LR as a function of token horizon for the 1.3B and 2.7B models. A straight-line provides a good fit, indicating that the power-law of Equation (1) works well. See Section 3.2 for further details.

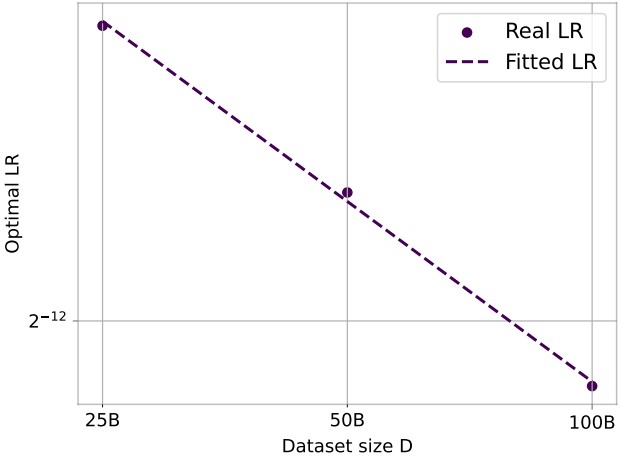

Figure 14: Optimal LR vs token horizon for a $7B$ LLama-1 model using batch size of $4M$.

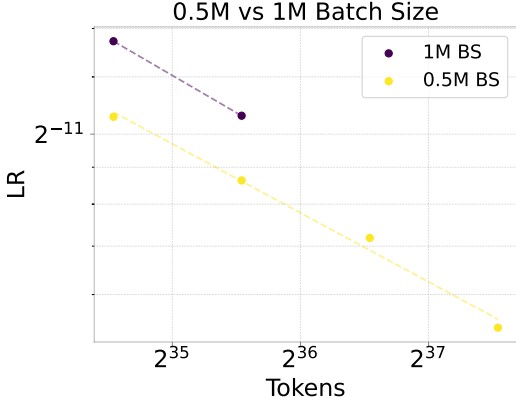

Figure 15: Optimal LR as a function of token horizon for a 1.3B model using a batch size of 0.5m or 1m tokens. A larger batch size implies that a larger LR is optimal, as expected. The two lines are roughly parallel, suggesting that the dependence on token horizons is the same irrespective of the batch size.

| Token Horizon | Optimal LR |
|---|---|
| 25B | $3.4 \times 10^{-4}$ |
| 50B | $2.8 \times 10^{-4}$ |
| 100B | $2.3 \times 10^{-4}$ |
| 1T (Using Eq. 1 | $1.15 \times 10^{-4}$ |
| 1T (Using Eq. 3) | $1.1 \times 10^{-4}$ |

Table 8: Optimal LR for LLama-1 for different token horizon using the fit in Figure 14. Note that the real Llama-1 used an LR of $3 \times 10^{-4}$ which our results suggest is significantly too large.

| | 25B | 50B | 100B | 200B | 400B | 800B |
|---|---|---|---|---|---|---|
| Optimal LR | $1.34 \times 10^{-3}$ | $1.02 \times 10^{-3}$ | $6.60 \times 10^{-4}$ | $4.12 \times 10^{-4}$ | $2.51 \times 10^{-4}$ | $1.98 \times 10^{-4}$ |
| Predicted LR | | | | $4.77 \times 10^{-4}$ | $3.35 \times 10^{-4}$ | $2.35 \times 10^{-4}$ |
| Ratio | | | | 0.864 | 0.749 | 0.843 |

Table 9: Simulated hyperparameter transfer across token horizons for a 125m model. The setup follows Table 9.

| | $R^2$ | $\beta$ |
|---|---|---|
| $\mu$ | 0.961 | 0.384 |
| $\sigma$ | $8.85 \times 10^{-3}$ | $7.78 \times 10^{-3}$ |
| $\sigma/\mu$ | $9.21 \times 10^{-3}$ | $2.03 \times 10^{-2}$ |

Table 10: We show the mean $\mu$, standard deviation $\sigma$ and relative deviation $\sigma/|\mu|$ estimated via bootstrapping for two quantities : the $R^2$ and $\beta$ in Section 3.4. The relative deviation of $\beta$ is roughly 10%, demonstrating that the uncertainty is reasonably small in our scaling laws.

## C  DERIVATION

In section 3.2 we saw empirically that for a given $N$ and constants $B, \beta$ that may be dependent on $N$,

$$LR^*(N, D) = B(N)D^{-\beta(N)} \tag{6}$$

As seen in Figure 6b, we observe a linear relationship between $\log N$ and $LR^*$ for any given $D$. This suggests the relationship:

$$LR^*(N, D) = A(D)N^{-\alpha(D)} \tag{7}$$

The constants $A, \alpha$ may depend on $D$. The key question that arises is how the general notion of model size $N$ can be incorporated into the joint scaling law. Moreover, the scaling law formula from Eq. 7 for constant $D$ has to be representable by Eq. 6. It is anticipated to align with the latter, consisting of distinct power laws, each with specific parameters for different $N$ and $D$ values. Consequently, the objective is to identify a function that fulfills these criteria

$$LR^*(N, D) = B(N)D^{-\beta(N)} = A(D)N^{-\alpha(D)} \tag{8}$$

**Dataset size $D$ for different Model Size $N$.** As seen in Figure 6a, since the lines are parallel for any given $N$, the slope $\beta$ (Eq. 6) is independent of the model size $N$. Therefore we can assume that $\beta(N) = \beta$.

**Model size $N$ for different Dataset size $D$.** As seen in Figure 6b, since the lines are parallel for any given $D$, the slope $\alpha$ (Eq. 7) is independent of the dataset size $D$. Therefore we can assume that $\alpha(D) = \alpha$ is constant.

Using the fact that $\alpha, \beta$ are constant and multiplying Eq. 8 by $N^\alpha D^\beta$:

$$B(N)N^\alpha = A(D)D^\beta \tag{9}$$

The LHS of Eq. 9 only depends on $N$, whereas the RHS only depends on $D$ so they should both equal some constant, $C$ (this step relies on our proof above that $\alpha, \beta$ are independent of $N, D$), resulting in the functional forms of $A(D), B(N)$

$$A(D) = CD^{-\beta}, B(N) = CN^{-\alpha} \tag{10}$$

Plugging the functional forms of $B(N)$ we finally get the final functional form for the joint scaling law as in Eq. 3.

