# OpenReview forum: "Scaling Optimal LR Across Token Horizons"
_ICLR.cc/2025/Conference — ICLR 2025 Poster_

### Official Review · Reviewer_VnBU · 2024-10-17

**Soundness:** 3
**Presentation:** 3
**Contribution:** 2
**Rating:** 6
**Confidence:** 3

**Summary:**

The paper shows that the optimal learning rate is decreasing in dataset size across scaling parameterizations, in contrast to most existing work on scaling, which studies model size (depth/width) only, and not data budget. They find that the optimal learning rate decreases approximately as a power law in the data budget, and provide a heuristic to help practitioners select an optimal learning rate scaling over data budgets. Their experiments include language model pretraining runs up to 2.7B parameters and 800B tokens, on which they fit these scaling trends.

**Strengths:**

I have not seen works touching on optimal LR with respect to data budget before, so this is neat. The message of the paper is clean and simple: optimal learning rate for language model pretraining decreases approximately as a power law in the data budget. In particular some strength are:

- The message is short and simple, and the empirics are done with reasonably standard architectural decisions, and are -- most importantly -- at large scale. I imagine this was a reasonably expensive paper to write in terms of compute.
- They check multiple parameterizations, including $\mu P$, which is important.
- The validate their predictions and include a nice touch of showing an actionable consequence: that the LR of Llama was not tuned optimally.
- The batch size factoring in the optimal learning rate is neat and intuitive and a good sanity check since the BS and LR are often optimized/tuned together in practice.

**Weaknesses:**

My gripes are mostly methodological.

- In my mind, the main contribution is an empirical one: the dots you plot. I didn't put much weight into the actual fitted scaling laws because you fit a *separate set of constants to each curve* as you varied token horizons for a fixed model size. As I understand it, the main point of a scaling law is that you should be fitting a *fixed set of constants for all token horizons* and THAT is what you should be plotting. I suspect this is the reason for your unusually strong fit. The quote from von Neumann comes to mind: "With four parameters I can fit an elephant, and with five I can make him wiggle his trunk." Ideally, you should even be using one fitted set of constants ACROSS model sizes, but since your focus is on data and not model size scaling, I can forgive this. But one set of fitted constants per model size is a bare minimum: the current setup for how fitting is done is misleading if I am correctly understanding it.

- It is unclear whether the main (scaling) experiments are done on the absolutely right type of architecture, ie. a modern "Transformer++," which is what they'd need to be to be most relevant to practice. Using architectural and hyperparameter choices from GPT3 is suboptimal because optimized Transformer++ architectures today use things like RMSnorm, RoPE embeddings, no linear biases, Adam $\beta$ values of $(0.9, 0.95)$, etc. Most people who work on pretraining know this, and the GPT3 architecture is certainly far from an optimized modern (2024) version by most people's standards. I see there are some preliminary experiments with the "Llama-1" architecture (though I'm not sure if it includes all the elements I outlined above). I believe the results, to be clear, and am not asking for new experiments, I just think there are some strange architectural choices even if mostly they are standard. Maybe including one ablation sweep with the architecture I described above (see perhaps the OLMo architecture for an example of a vanilla Transformer++) to check the same trends hold in the same way.

- The functional forms you posit and fit are somewhat arbitrary. You choose a quadratic and power law, respectively, but do not explore other fits as I understand it, or even justify from any theoretical perspective why these should be the correct fits. Again, to be clear, I personally think these posited forms are fine, but they are indeed arbitrary and this requires justification.

- You definitely need to ablate LR schedule to make sure these are not an artifact of cosine LR schedule.

**Questions:**

- If my understanding is correct, $\mu P$ describes a particular scaling scheme for how to adjust hypers as you scale width or depth (the latter being described in more recent depth-$\mu P$ extensions, like [1]). When you say you "scale in $\mu P$" but then sweep tokens at a *fixed* model size, I don't know what this means, or how it's different from standard parameterizations? Are you just referring to using a particular initialization scale, then? Can you explicitly tell me how you are scaling in the main (standard) vs $\mu P$ experiments and how they differ if you are not varying model size in the latter, for instance in Figure 5?
- It is known why we need to decrease LR with model size, $N$, and it has to do with reasons relating to how EoS/sharpness scale in $N$, for instance see [2]. Is there a similar theoretical reason why the optimal LR should be smaller on larger data budgets? Why should we expect a priori lower LR* when training for longer (especially since LR schedules usually decrease LR over training anyway)?

[1] Bordelon, Blake, et al. "Depthwise hyperparameter transfer in residual networks: Dynamics and scaling limit." arXiv preprint arXiv:2309.16620 (2023).

[2] Noci, Lorenzo, et al. "Why do Learning Rates Transfer? Reconciling Optimization and Scaling Limits for Deep Learning." arXiv preprint arXiv:2402.17457 (2024).

---

> ### Author Response · Authors · 2024-11-30
> **Thanks for your comments**
>
> We thank the reviewer for his/her extensive comments and great feedback.  We are happy that the reviewer finds our work novel and impactful, writing `I have not seen works touching on optimal LR with respect to data budget before, so this is neat`. We have completed an additional experiment to address concerns raised by the reviewer.  We've suffered an electrical outage in the area, this has delayed the rebuttal which we apologize for.  Below we answer individual questions.
>
> Q1: ```It is unclear whether the main (scaling) experiments are done on the absolutely right type of architecture, ie. a modern "Transformer++," ```
>
> A1: This is a great point! We have added an experiment using the Llama architecture to the Appendix as Figure 11. We find very similar scaling laws as for the GPT-3 architecture. We wanted to mimic the GPT-3 setup to be “traditional”. Note that the 7B model of section 4.1 uses the Llama architecture.
>
> Q2: ```The quote from von Neumann comes to mind: "With four parameters I can fit an elephant, and with five I can make him wiggle his trunk.```
>
> A2: We appreciate the comment but believe that we are not “overfitting” when fitting our scaling laws. Both Table 1 and Figure 7 use a held-out validation set to evaluate the generalization of our scaling laws. Furthermore, typically use 2 parameters (3 in the case of eq 3), linear models, and always have more data points than free parameters. But we have clarified this point further in the paper.
>
> Q3: ``` one set of fitted constants per model size is a bare minimum```
>
> A3: We agree, and this is the protocol we follow. See e.g. Figure 4 where each model has up to 6 data points, and we fit two parameters (beta and D in equation 1) for each model. In Figure 7 we fit three parameters (C alpha and beta in equation 3) across multiple models. Is there some kind of misunderstanding here?
>
> Q4: ```The functional forms you posit and fit are somewhat arbitrary. [...], I personally think these posited forms are fine, but they are indeed arbitrary and this requires justification.```
>
> A4: This is a good point! We have added motivation for the forms we use.
>
> Q5: ```You definitely need to ablate LR schedule to make sure these are not an artifact of cosine LR schedule.```
>
> A5: We have been asked to conduct more experiments which we can. We have prioritized completing the experiments with the Transformer++ model you suggested, and also experiments on duplicate tokens requested by reviewer ikK6. Cosine is a very standard LR schedule used e.g. in Llama-3, and we believe that it is representative of what is commonly used in practice. For computational reasons, we must defer this question to future work.
>
> Q6: ```If my understanding is correct, μP describes a particular scaling scheme for how to adjust hypers as you scale width or depth (the latter being described in more recent depth-μP extensions, like [1]). When you say you "scale in μP" but then sweep tokens at a fixed model size, I don't know what this means, or how it's different from standard parameterizations? Are you just referring to using a particular initialization scale, then? Can you explicitly tell me how you are scaling in the main (standard) vs μP experiments and how they differ if you are not varying model size in the latter, for instance in Figure 5?```
>
> A6 You are correct, usually muP includes muTransfer over width. In this case, we indeed use a fixed model size. Meaning we change the initialization and the attention scaling by d instead of sqrt(d).  muP was not designed to transfer across token horizons, we just wanted to experimentally verify that it didn’t.
>
> Q7: ```Is there a similar theoretical reason why the optimal LR should be smaller on larger data budgets?```
>
> A7: This is a fair point. Our paper is mostly empirical and does not provide such an explanation. We follow a tradition of impactful and purely empirical like Chinchilla which also didn’t provide much theoretical justification. Providing theoretical motivation is important, but we will defer a more theoretical study to future work.
>
> We have run an experiment to address the concern about the tranformer++ model, and we have also clarified that some of the other requests are considered out of scope for computational reasons, would you consider increasing your score? If not, is there any other changes we can make to improve the paper?

---

### Official Review · Reviewer_yU6o · 2024-11-03

**Soundness:** 3
**Presentation:** 3
**Contribution:** 2
**Rating:** 6
**Confidence:** 3

**Summary:**

The paper conducts an empirical study to characterize scaling law of optimal step-size with respect to the token used in the training of LLMs. They make some interesting observations: 1- best step-size shrinks approximately according to D^.32 where D is the training horizon and 2- this scaling law is almost consistent for different network sizes.

**Strengths:**

The paper is clear and well written. It delivers what it promises.

This is an impactful research area. Efficient methods or scaling laws for finding best step-sizes helps to improve overall training efficiency and performance.

**Weaknesses:**

The paper provides no analysis or in-depth intuition on why optimal LR scales with exponent -.32 ~= -1/3.

The scale of experiments are rather small for a fully empirical paper. Unless there is an analytical explanation or intuition provided for the observed patters, the trends might be unreliable for larger networks and longer horizons. In the same vein, would the scaling law remain the same for other transformers like Llama and Mistral? It would also be interesting to study how optimal LR scales with respect to training horizon in other architectures like ResNet and MLP.

In general, this is a solid paper, however in my view it falls marginally below the ICLR bar in terms of contributions. See below for some suggestions that I think would help in improving the contributions and the scores.

**Questions:**

Can the authors provide any theoretical analysis or intuition for why the optimal learning rate scales with an exponent of approximately -1/3?

Could the authors use their fitted scaling law to propose a new LR schedule. For example, an LR schedule that decays with t^{-.32}. Does this outperform cosine decay? Is -.32 the best decay exponent for LR schedule?

Could you strengthen your claims by testing the scaling law on other transformer architectures like Llama and Mistral, as well as non-transformer architectures like ResNets and MLPs? This would help demonstrate the generality of the observed scaling behaviour.

---

> ### Author Response · Authors · 2024-11-29
> **Thanks for your comments**
>
> We thank the reviewer for his/her great feedback, and are especially pleased that the paper is considered `clear and well written` and that it is `an impactful research area`. We have added an experiment to address concerns raised by this reviewer.  We've suffered an electrical outage in the area, this has delayed the rebuttal which we apologize for.  Below we address individual points raised in the review.
>
> Q1: ```Could you strengthen your claims by testing the scaling law on other transformer architectures like Llama and Mistral```
>
> A1: This is a great suggestion, thanks! We have added an experiment (using the Llama architecture) as Figure 11 in the paper. We see similar trends and scaling laws for this model architecture. Also, note that Figure 7 already uses the Llama architecture for the 7B model, we have clarified this in the caption.
>
> Q2: ```The scale of experiments are rather small for a fully empirical paper. ```
>
> A2: We respectfully disagree with this. Let us consider Figure 2 and use the approximation that FLOPs equals parameters times tokens. The total FLOPs then end up as ~2.4e21. That’s just one figure – we replicate the same experiment across many models and provide many ablations. We consider many model sizes and additionally provide ablations. Note that reviewer VnBU writes that `I imagine this was a reasonably expensive paper to write in terms of compute.`. Similarly, reviews ikK6 writes that `The paper has significant strong points. 1.Experimental Scale. The authors had the capability to run large-scale experiments (billion scale). This makes the results very reliable in the specific experimental setting adopted here (i.e. architecture, model size, optimizer setting)`.
>
> Q3: ```The paper provides no analysis or in-depth intuition on why optimal LR scales with exponent -.32 ~= -1/3.```
>
> A3: This is a fair point. Our paper is mostly empirical, and the exact exponents are typically hard to motivate theoretically without very strong assumptions. We follow a tradition of impactful and purely empirical like Chinchilla which also didn’t provide much theoretical justification. Providing theoretical motivation is important, but not the focus of this paper.
>
> Q4: ```Could the authors use their fitted scaling law to propose a new LR schedule. ```
>
> A4: This is a good point and a great direction for follow-up work! Unfortunately, we have only used the standard cosine, and can not draw conclusions about new LR schedulers. Not ablating the LR schedule has been a conscious choice to limit the computational requirements of our study. We have clarified this in section 6. Thus, we must defer this question to future work.
>
> Given that we have run the experiment the reviewer requested, and that we have clarified that some of the other requests are considered out of scope for computational reasons, would you consider increasing your score? If not, is there any other changes we can make to improve the paper?

---

> ### Comment · Reviewer_yU6o · 2024-12-01
> **After Reading Authors Response**
>
> After reviewing the authors' responses and the updated manuscript, my concerns have been partially addressed. However, the experimental scope remains limited relative to the broader scale of contemporary pretraining practices for LLMs. For instance, the use of the Llama-7B model with only 100B tokens represents a shorter horizon compared to current standard practices in LLM pretraining.
>
> The authors note that their work is fully empirical and that theoretical analysis is beyond the scope of this study. While this approach is reasonable, the absence of theoretical or intuitive justification limits confidence in the scaling law's applicability to longer horizons or larger models. Nonetheless, the observation that optimal lr strongly depends on the horizon in a particular way is an interesting contribution and may inspire future research to build upon this foundation. On this grounds, I raise my score from 5 to 6.

---

### Official Review · Reviewer_8GV3 · 2024-11-05

**Soundness:** 3
**Presentation:** 3
**Contribution:** 3
**Rating:** 6
**Confidence:** 4

**Summary:**

As LLMs are scaled up it is not possible to tune hyperparameters at the largest scale. This necessitates a way of predicting optimal hyperparameters. This paper focuses on predicting optimal LR as the training dataset size is scaled. Specifically, they fit a scaling law for optimal LR as a function of dataset size and find 1. Good fits to the scaling law and 2. The exponent is negative i.e. optimal LR decreases with increasing tokens. They also find that they can fit the loss vs log-LR curve by a quadratic (At least around the optimum value).

**Strengths:**

As described, the problem studied by the paper is important and the paper's findings give an important starting point for predicting optimal LR.

**Weaknesses:**

There are some other hyperparameters which strongly interact with learning rate such as weight decay and warmup. The paper does not explore the interaction between these factors and learning rate.

**Questions:**

The warmup used by the authors is much smaller than that used in many recent works[1].


To confirm that the results are robust to values of other hyperparameters, could the authors report results on one of the setups but with 10% warmup steps and 0 weight decay? Assuming the results are robust, I would be happy to increase my score.


[1] SMALL-SCALE PROXIES FOR LARGE-SCALE TRANS-
FORMER TRAINING INSTABILITIES.

---

> ### Author Response · Authors · 2024-11-29
> **Thanks for your comments**
>
> We thank the reviewer for his/her comments, questions, and suggestions for improving our paper!  We are excited that the reviewer believes that `the problem studied by the paper is important`. Note that we've suffered an electrical outage in the area, this has delayed the rebuttal which we apologize for. We answer individual points below:
>
>
> Q1: ```There are some other hyperparameters which strongly interact with learning rate such as weight decay and warmup. The paper does not explore the interaction between these factors and learning rate.```
>
> A1: This is a great point! Given our limited computational resources, we haven’t been able to fully explore interactions with all hyperparameters. Indeed, WD is probably the most salient hyperparameter here and we consider this a good topic for follow-up work. We have added further discussion regarding this in section 6!
>
> Q2: ```The warmup used by the authors is much smaller than that used in many recent works[1].```
>
> A2: Thanks for highlighting this! We have generally followed the practice of Muennighoff et. al. and used 1 % of the total steps for warmup. As per results from Wortsman et. al, too short a warmup can cause instabilities. Hence, we chose to cap the warmup to 1000 steps minimum. Note that e.g. Llama-1 uses 2k warm-up steps. As per Figure 5 in Wortsman et. al., the worst instabilities are for 500 or fewer steps. We don’t see much instability in our experiments, so we believe that our warmup is sufficient.
>
> Q3: ```could the authors report results on one of the setups but with 10% warmup steps and 0 weight decay?```
>
> A3: This is a great suggestion! We’ve only been able to run two requested experiments – see the added Figures 11 and 12. We have added this experiment to our todo list, but have not been able to complete it yet.
>
> We have run an experiment to address the concern about the tranformer++ model, and we have also clarified that some of the other requests are considered out of scope for computational reasons, would you consider increasing your score? If not, is there any other changes we can make to improve the paper which do not require additional experiments?

---

### Official Review · Reviewer_ikK6 · 2024-11-05

**Soundness:** 3
**Presentation:** 3
**Contribution:** 2
**Rating:** 6
**Confidence:** 4

**Summary:**

The paper studies how the optimal learning rate changes when varying the total number of tokens that are fed into GPT-3-like models (which, equivalently, means training for a longer time). Consistently, it is found that increasing the number of tokens corresponds to a decrease in the optimal learning rate. This inverse relationship is then studied more in detail by fitting a power law, and its extrapolation allows for a precise prescription of how the learning rate should scale with the number of tokens. Finally, this approach has discovered that LLAMA-1 has been potentially trained with a too-large learning rate.

**Strengths:**

The paper has significant strong points.

1. Experimental Scale. The authors had the capability to run large-scale experiments (billion scale). This makes the results very reliable in the specific experimental setting adopted here (i.e. architecture, model size, optimizer setting).

2.  The research question of how the learning rate should scale up is very important in practical settings and it is not covered in either theoretical or empirical research. Thus, the experimental findings are both novel and relevant.

3. The case study on LLAMA training is very interesting, for instance, by retrospectively questioning how hyperparameter tuning was performed in that setting.

**Weaknesses:**

The paper has some weaknesses, mainly due to the depth of the investigation that is performed. More concretely (in order of importance):

1. It is unclear whether the observed inverse relationship between the number of tokens used for pretraining and optimal learning rate is due to the fact the model is trained progressively for a longer number of time steps, or because the network has processed more data. This is quite a fundamental experiment, and it’s unclear what the view taken by the authors is, given that the meaning of the term “token horizon” is not entirely specified. This could be tested, for instance, by fixing the amount of data and training for multiple epochs, and by increasing the batch size while fixing the number of steps (this is already partially done in the current paper). Aggregating these experiments together with the ones already performed by the authors should elucidate the aforementioned question.

2. The experiment of Section 4 ignores the fact that as the scale gets larger, the value of $\beta$ is very different. Thus, with the provided evidence these results are hardly predictive of the optimal learning rate in the joint scaling (model size, token horizon), and more investigation is needed. In fact, the observation that $\beta$ changes with scales (which the authors make) should already advise against the approach of independently fitting the constants for model size and token horizon. However, the authors explicitly advertise this suggestion to practitioners. On the other hand, it would have been more appropriate to see reported how $\beta$ changes with scale, especially for $\mu$P. In particular, for $\mu$P we expect the learning rate to transfer across width, so the joint scaling properties should be more feasible to test.

3. The authors observe a different $\beta$ for different model size (Table 5). Fundamentally, the authors increase the model size by increasing both the width and at times the depth of the model. Thus, it is unclear whether the observed different $\beta$ at different model sizes is due to the width or the depth scaling.


Minor:

4. Another fundamental limitation of this work is that $\beta$ would potentially change as any architectural modification is made (e.g. QK norm, attention method, gating, etc..). Thus, the $\beta=0.32$ proposed to practitioners, on top of the problems stated above, is valid only in the GPT-3 setting studied. I think this should be emphasized more to not be misleading.

5. I am not entirely clear on the purpose of the $\mu$P  experiment. $\mu$P is just not designed to exhibit hp transfer across a number of samples (in this case, the number of tokens, or equivalently training time). Thus it is not entirely clear what the expectation of this experiment was in the first place. However, in the related work section, the authors state that exploring this limitation was part of the objectives of the paper, hence I do not decrease my score for this reason.

6. The $\mu$P extension to depth scaling is derived in at least two existing works [1,2]. Thus, there is a parametrization that exhibits learning rate transfer across depth as well. However, I would partially still agree with the authors that the scaling with respect to depth still has to be fitted with power laws (as it is advertised to practitioners), due to the nature of Transformers that have >1 layers within a single residual block. Despite this, I would appreciate it if a more thorough/informed explanation of these suggestions were present in the paper.

[1] Tensor Programs VI: Feature Learning in Infinite-Depth Neural Networks

[2] Depthwise Hyperparameter Transfer in Residual Networks: Dynamics and Scaling Limit

I am willing to raise my score if my concerns are resolved.

**Questions:**

What do these results seem to suggest about a possible “$\mu$P extension” to a simultaneous scaling limit of the multiple dimensions (token, width)?

**Update**

I thank the authors for their additional answers.

I apologize if it was not very clear, but please let me stress the fact that my point about the semantics of what token horizon means underlies deep questions about the paper's claim and the extension of their validity. In my honest opinion, this paper presents exciting results about how the optimal learning rate evolves with more training time. However, it fails to define adequate boundaries for the (empirical) claims. In this respect, the fact that no specific meaning is attributed to token horizon stems from these missing experiments that would help to disentangle the various causes of the right shift of the optimal learning rate. Therefore I would not say that the new results are *"completely consistent with our previous experiments"*: however, they are consistent with the revised notion of token horizon that does **not** equal dataset size. And I do sincerely appreciate the authors for promising to update the paper, and carefully revising the notion of token horizon across the manuscript.

The fact that $\beta=0.32$ transfers across architectures is itself a very counterintuitive claim, and in my opinion, it would require significantly more ablations to be verified. I would at least state that it could be that more investigation is needed for architectures that are not GPT or Llama-1, and $\beta=0.32$ might not be the right scaling exponent outside the tested framework (I thank the authors for pointing out Figure 11). At a fundamental level, no explanation is provided as to what and why the shift happens. Thus, I think these limitations have to be addressed (i.e. stated).

I am referring to the papers that extend the lr transfer to the network's depth: https://openreview.net/forum?id=17pVDnpwwl and https://arxiv.org/abs/2309.16620.

Overall, I think that with the agreed revision of the storyline around the conceptualization of token horizon, and with the additional experiments, this paper deserves a slightly higher score. Thus, I am updating it to 6.

---

> ### Author Response · Authors · 2024-11-30
> **Thanks for your comments**
>
> We thank the reviewer for the throughout and constructive review. We are especially happy that the reviewer thought that `the experimental findings are both novel and relevant`. We have run two additional experiments requested by the reviewer. We largely agree with the reviewers that there are many open questions that our paper does not answer – this is an unfortunate necessity given our limited computational resources. We intend to answer a few questions and would hope that this could inspire follow-up work to answer more. Note that we've suffered an electrical outage in the area, this has delayed the rebuttal which we apologize for. With that said, we now address individual points raised by the reviewer and clarify a few likely misunderstandings.
>
> Q1: ```It is unclear whether the observed inverse relationship between the number of tokens used for pretraining and optimal learning rate is due to the fact the model is trained progressively for a longer number of time steps, or because the network has processed more data. [...] This could be tested, for instance, by fixing the amount of data and training for multiple epoch,. ```
>
> A1: This is a great point! By token horizon we do not mean unique tokens – so two epochs over a fixed dataset would double the token horizon. We have run the experiment suggested by the reviewer, see Figure 12 in the Appendix. We sample 25B unique tokens and consider multiple epochs. We see that the scaling looks very similar to 600B unique tokens. After the PDF deadline will additionally complete experiments with 4 epochs, and they look the same, see [here](https://imgur.com/gallery/images-0to7u0X). Muennighoff et. al. show that LLMs see diminishing returns after ~4 epochs, so (within a reasonable number of epochs) the total token horizon is responsible for the scaling we see, not the number of unique tokens.
>
> Q2: ```Another fundamental limitation of this work is that would potentially change as any architectural modification is made (e.g. QK norm, attention method, gating, etc..). Thus, the proposed to practitioners, on top of the problems stated above, is valid only in the GPT-3 setting studied.```
>
> A2: Please note that the original submission did not only consider the GPT-3 model, the 7B model in 7 uses the Llama architecture. We have clarified this in the caption. Furthermore, in section 4.1 where we write that “ we adopt the LLama-1 architecture (RMSnorm, Rope embeddings, and so on) and run small-scale experiments with token horizons 25B, 50B, and 100B and different LRs.”. We have additionally run more experiments with the Llama-1 architecture and added these as Figure 11 in the paper. Thus, we consider both the GPT-3 and Llama architecture. However, the general point still holds that beta conceivably could change with model architecture. Figure 7 demonstrates generalization across the two architectures we consider. Doing ablation over more architectures, like depth vs width, is left for future work due to our computing constraints.
>
> Q3: ```the provided evidence these results are hardly predictive of the optimal learning rate in the joint scaling (model size, token horizon)```
>
> A3: We believe that our results are predictive for joint scaling. See e.g. Figure 7 and note that the caption states that “The data points for the 7B model of Section 4.1 are excluded at the time of fitting and used as validation data.”. We specifically do this to validate that our laws are predictive. Table 4 similarly uses held-out data. We have clarified this point in the paper further.
>
> Q4: ```The experiment of Section 4 ignores the fact that as the scale gets larger, the value of β  is very different. Thus, with the provided evidence these results are hardly predictive of the optimal learning rate in the joint scaling (model size, token horizon), and more investigation is needed. In fact, the observation that β changes with scales (which the authors make) should already advise against the approach of independently fitting the constants for model size and token horizon. However, the authors explicitly advertise this suggestion to practitioners. ```
>
> A4: This is likely a misunderstanding, we do not ignore this. Our experiments show that while Beta varies with small model sizes, it asymptotes for larger model sizes. This is demonstrated in Figure 6, where the same Beta is used for all models and we still see a good fit. Indeed, we write that `For practitioners who are working on larger models (say >= 760m) we recommend simply using Equation (3) where we have already found β = 0.32 to generalize`. Please note that we don’t make this recommendation for smaller models.

---

> ### Author Response · Authors · 2024-11-30
> **part 2**
>
> Q5: ```I am not entirely clear on the purpose of the muP experiment.```
>
> A5: Thanks for this feedback! We wanted to demonstrate that optimal LR changes with token horizon even when muP is used. As you state, muP was not designed to transfer this way. We just wanted to experimentally verify that it didn’t. Note that reviewer VnBU writes that `They check multiple parameterizations, including muP, which is important.`.
>
> Q6: ```Thus, it is unclear whether the observed different at different model sizes is due to the width or the depth scaling.```
>
> A6: This is a fair point! To limit computational needs, we have specifically not focused on model architecture. We believe that this is an interesting question that could be answered in follow-up work. We have clarified this.
>
> Q7: ```The P extension to depth scaling is derived in at least two existing works [1,2]. [….] I would appreciate it if a more thorough/informed explanation of these suggestions were present in the paper.```
>
> A7: Thanks for this feedback! Equation (3) is LR*=C*N^(-a)*D^(-b). We know from https://arxiv.org/abs/2001.08361 that N=C’ * n_layer * d^2 for some constant C’. Plugging this in (3) we get: LR*=C*C’^(-a)n_layer^(-a)*d^(-2a)*D^(-b). Since LR* is independent when using muP, we know we can ignore it and if we define a new constant c:=C*C^(-a) we finally get LR*=c*n_layer^(-a)*D^(-b) which is the final equation we have in the paper. Do you want us to add this derivation in the main text? It was not retained to keep the paper concise.
>
> Q8: ```What do these results seem to suggest about a possible “μP extension” to a simultaneous scaling limit of the multiple dimensions (token, width)?```
>
> A8: Does the answer to Q7 suffice for this question? If not, we are happy to elaborate.
>
> Given that we have run two experiments requested by the reviewer, and made numerous other clarifications -- would the reviewer consider increasing the score? If not, is there anything else we can do? Thanks!

---

> > ### Comment · Reviewer_ikK6 · 2024-12-02
> > **Response to Rebuttal**
> >
> > I sincerely thank the authors for their response, and I am sorry to hear that they had trouble with power outages.
> >
> > Q1: I thank the authors for the clarification and the additional experiment. Upon re-reading the paper in light of these clarifications, the setup is still misleading. I think what the authors intend as "dataset size" is crucial here, and not to be confused with training steps.  The new experiment suggests that actually, the shift to the right of the optimal learning rate happens (with roughly the same $\beta$) when the *dataset size is fixed*, and an extra epoch is performed.
> >
> > Q2: I thank the authors for the explanation and for including more details in the caption. The experiment that the optimal $\beta \approx 0.32$ transfers across architectures (i.e. from GPT-3 to Llama) is very counterintuitive and I feel that it should be more thoroughly investigated. It seems to suggest that the optimal $\beta$  is not determined by the architecture, but only by the data. In this setting, it would have been nice to try different datasets to see what factors influence $\beta$ (the current paper only tests $\beta$ on RefinedWeb).
> >
> > Q3-Q4: Again, the fact that the architecture has changed makes it hard to establish whether the transfer from small to large scale happens because of architectural changes *combined with* scale, instead of entirely from scale. The same could be said for much longer training horizons, that are more in line with practice.
> >
> > The current section on advice for practitioners proposed a $\beta=0.32$, regardless of the dataset or architecture, which is misleading without further evidence.
> >
> > Overall, I think the paper would have been significantly more comprehensive if the authors had investigated what factors do and do not influence the optimal scaling exponent, trying to disentangle them with proper ablations. This would have confined (or extended) the region of validity of the presented findings. In the current form, the role of architecture, and the number of data points vs training steps require more investigation. Also, the new evidence suggests that the token horizon does not equal dataset size, as the authors defined in the abstract. Finally, I would mention in the same section that the transfer has been analyzed in a few papers already.

---

> ### Author Response · Authors · 2024-12-03
> **Thanks for your comments**
>
> Thanks for your comments! We agree with you that we have been less formal and consistent with the term "token horizon" than what is needed. We can commit to updating the paper to clarify that "token horizon" is the total number of tokens seen during training in the sense that doubling the number of epochs doubles the token horizon. Both ChatGPT and perplexity seems to believe that `training tokens` or `token count` as apt terms, see https://imgur.com/a/tokens-taOhnD3. We will update the abstract to read:
>
> ```
> State-of-the-art LLMs are powered by scaling – scaling model size, total token count, and cluster size. It is economically infeasible to extensively tune hyperparameters for the largest runs. Instead, approximately optimal hyperparameters must be inferred or transferred from smaller experiments. Hyperparameter transfer across model sizes has been studied in Yang et al. (2022). However, hyperparameter transfer across training tokens – or token horizon – has not been studied yet....
> ```
> Given this clarification and commitment to more clearly define the term token horizon, and the fact that we have run two additional experiments specifically requested by the reviewer -- would you consider increasing your score? We sincerely hope that the reviewer won't give the paper a low score just because of the semantics of the term "token horizon". The electrical outages unfortunately delayed our plans and we weren't able to update the PDF with this which we apologize for. Below we address individual points raised:
>
>
> Q1: ```what the authors intend as "dataset size" is crucial here, and not to be confused with training steps.```
>
> A1: We agree, and we earlier clarified that `By token horizon we do not mean unique tokens – so two epochs over a fixed dataset would double the token horizon` in an earlier comment. We believe that referring to the total number of tokens seen during training as token horizon is standard terminology, but we will clarify this in the abstract and in the main text. Both chatGPT and perplexity seems to thing that training tokens or total token count is an apt term -- see https://imgur.com/a/tokens-taOhnD3.  Thanks for highlighting this!
>
> Q2: ```The new experiment suggests that actually, the shift to the right of the optimal learning rate happens (with roughly the same
> ) when the dataset size is fixed, and an extra epoch is performed.```
>
> A2: Yes, and this is completely consistent with our previous experiments and interpretations that the total token horizon is what matters. We never made any claims that the number of unique tokens was determining the optimal LR.
>
> Q3: `Upon re-reading the paper in light of these clarifications, the setup is still misleading.`
>
> A3: We are happy to clarify the semantics of the term "token horizon" further in the paper. We believe that token horizon is a standard term used when training industry-strength LLMs, which we have done in separate works. Though we hope that the reviewer won't give the paper a low score just for the sake of semantics for this term.
>
> Q4: `I feel that [different model architectures] should be more thoroughly investigated. ... it would have been nice to try different datasets to see what factors influence beta`
>
> A4: We have intentionally focused on LR and token horizon, and only provide small-scale experiments for other factors. This is intentional, and not providing large-scale ablations on datasets or architecture is a conscious choice to limit the compute requirements. We have already run two additional experiments which the reviewer has requested, and believe that these questions are a good fit for future work.
>
> Q5: `Finally, I would mention in the same section that the transfer has been analyzed in a few papers already.`
>
> A5: Thanks for bringing this to our attention. Could the reviewer provide the specific papers he/she is referring to here? We will cite them.
>
> Q6: `The current section on advice for practitioners proposed a, regardless of the dataset or architecture, which is misleading without further evidence.`
>
> A6: This is a fair point -- we are happy to update the paper to mention that we haven't ablated datasets. But we have tested different architectures and model sizes, so we respectfully disagree that our claim is misleading with respect to architectures.
>
> Q7: `Again, the fact that the architecture has changed makes it hard to establish whether the transfer from small to large scale happens because of architectural changes combined with scale, instead of entirely from scale. `
>
> A7: Please note that the Llama experiments we added to the appendix as Figure 11 used the same model size as the GPT-3 model. We there see transfer across model architectures when the model size is held constant. So we have controlled experiments which vary only the scale and only the model architecture.

---

### Public Comment · ~Laurence_Aitchison1 · 2024-11-20
**Prior work on hyperparameter transfer for dataset size / token horizon.**

This is a super-interesting paper, thanks to the authors!

As a quick note, there's prior work on hyperparameter transfer for dataset size:
https://arxiv.org/abs/2405.13698
It's a bit different, in that it argues that we should be modifying the weight decay, and not the learning rate as you change dataset size.  But it is clearly warrants considerable discussion in the present work.

---

> ### Author Response · Authors · 2024-11-29
>
> Thanks, we have cited the paper you linked in the updated PDF!

---

### Author Response · Authors · 2024-11-30
**Thanks for all the reviews**

We thank the reviewers for their interest and great feedback. We have updated the PDF with many clarifications and two additional experiments:

- Ablations using the Llama architecture, requested by multiple reviewers
- Ablations on multiple epochs, requested by reviewer ikK6

These are available in the appendix as Figures 11 and 12. We apologize for the late rebuttal. We've suffered an electrical outage in the area, this has delayed the rebuttal. We are happy to interact more with the reviewers until the deadline. We'd like to note that many reviewers have requested additional experiments we do not have the resources to run. The paper only studies two hyperparameters -- token horizon and LR. Adding additional experimental knobs -- such as LR schedule, WD hyperparameters, network depth -- is not feasible. We instead chose standard values for these following GPT-3 or Llama-1. We have made the conscious choice to limit the scope of the paper this way, as stated in section 6.

---

### Meta-Review · Area_Chair_tG8u · 2024-12-22

**Metareview:**

This paper studies the learning rate hyperparameter transfer problem across dataset sizes (i.e., token horizon) for large language models. The results show that the optimal learning rate becomes smaller as the token horizon length increases, and it follows a scaling law for any given architecture. This means it is possible to transfer the knowledge about optimal learning rate from shorter horizon to longer horizon. Interestingly the authors found that the learning rate used for LLamma-1 might be too large.

These are very interesting, novel and potentially impactful results. However, as the reviewers pointed out, the scope is very limited. Hyperparameters in the model training process are intertwined and can jointly affect the training results. Only focusing on learning rate can lead to biased conclusions. For example, it might be possible that the scaling law becomes different when other hyperparameters change values.

During discussion, the reviewers pointed out that "the paper is slightly superficial, in the sense that it does not analyze whether the source of the shift is purely training time or more data" and "this paper does not do a lot beyond the simple singular claim it puts in the title". But the reviewers also praised that the paper is interesting and necessary since "the current theory behind learning rate scaling does not consider time and data as scaling quantities", and "its topic is important and crucially the experiments are done in a large scale and controlled settings with a clear trend that persists under ablations", and the new experiments resolved some concerns about the source of the shift.

Overall I believe the contributions are valuable and the ICLR community can benefit from it. Hence I recommend acceptance.

**Additional Comments On Reviewer Discussion:**

During discussion, the reviewers pointed out that "the paper is slightly superficial, in the sense that it does not analyze whether the source of the shift is purely training time or more data" and "this paper does not do a lot beyond the simple singular claim it puts in the title". But the reviewers also praised that the paper is interesting and necessary since "the current theory behind learning rate scaling does not consider time and data as scaling quantities", and "its topic is important and crucially the experiments are done in a large scale and controlled settings with a clear trend that persists under ablations", and the new experiments resolved some concerns about the source of the shift.

Overall I believe the contributions are valuable and the ICLR community can benefit from it. Hence I recommend acceptance.

---

### Decision · Program_Chairs · 2025-01-22

Accept (Poster)